

# An Optimised OC/EC Fraction Separation Method for Radiocarbon Source Apportionment Applied to Low-Loaded Arctic Aerosol Filters

Martin Rauber[1,2], Gary Salazar[1,2], Karl Espen Yttri[3], Sönke Szidat[1,2]

[1]Department of Chemistry, Biochemistry and Pharmaceutical Sciences, University of Bern, Bern, Switzerland
[2]Oeschger Centre for Climate Change Research, University of Bern, Bern, Switzerland
[3]Department of Atmospheric and Climate Research, NILU – Norwegian Institute for Air Research, Kjeller, Norway

*Correspondence to*: Sönke Szidat (soenke.szidat@unibe.ch)

**Abstract.** Radiocarbon ($^{14}$C) analysis of carbonaceous aerosols is used for source apportionment, separating the carbon content into fossil vs. non-fossil origin, and is particularly useful when applied to subfractions of total carbon (TC), i.e., elemental carbon (EC), organic carbon (OC), water-soluble OC (WSOC), and water-insoluble OC (WINSOC). However, this requires an unbiased physical separation of these fractions, which is difficult to achieve. Separation of EC from OC using thermal-optical analysis (TOA) can cause EC loss during the OC removal step and form artificial EC from pyrolysis of OC (i.e., so-called charring), both distorting the $^{14}$C analysis of EC. Previous work showed that water extraction reduces charring. Here, we apply a new combination of a WSOC extraction and $^{14}$C analysis method with an optimised OC/EC separation that is coupled with a novel approach of thermal-desorption modelling for compensation of EC losses. As water-soluble components promote the formation of pyrolytic carbon, water extraction was used to minimise the charring artefact of EC, and the eluate subjected to chemical wet oxidation to $CO_2$ before direct $^{14}$C analysis in a gas-accepting accelerator mass spectrometer (AMS). This approach was applied to 13 aerosol filter samples collected at the Arctic Zeppelin Observatory (Svalbard) in 2017 and 2018, covering all seasons, which bear challenges for a simplified $^{14}$C source apportionment due to their low loading and the large portion of pyrolysable species. Our approach provided a mean EC yield of $0.87 \pm 0.07$ and reduced the charring to 6.5 % of the recovered EC amounts. The mean Fraction Modern ($F^{14}C$) over all seasons was $0.85 \pm 0.17$ for TC, $0.61 \pm 0.17$ and $0.66 \pm 0.16$ for EC before and after correction with the thermal-desorption model, respectively, and $0.81 \pm 0.20$ for WSOC.

## 1 Introduction

Considerable efforts have been made to investigate atmospheric aerosol due to its relevance on a wide range of environmental topics, including change of radiative forcing and adverse effect on human health (McNeill, 2017; Lelieveld et al., 2015; Landrigan, 2017; Pope et al., 2020). Exposure to ambient atmospheric particulate matter (PM) has been associated with damage to the cardiopulmonary system and causing at least 3 million premature deaths per year globally (Kim et al.,





2015; Lelieveld et al., 2015; Forouzanfar et al., 2016). Understanding aerosols is therefore crucial for future projections and for the improvement of air quality especially for severely affected areas (Quinn et al., 2008; Bond et al., 2013; Schmale et al., 2021). Although the Arctic is considered a pristine part of the world, it is also affected by emissions from polluted regions in the northern hemisphere, causing the Arctic haze phenomenon (Barrie, 1986; Heidam et al., 2004; Quinn et al., 2002; Zhao and Garrett, 2015; Engelmann et al., 2021; Jouan et al., 2014), occurring in late winter and early spring and have

been known for decades (Barrie et al., 1981). Arctic haze consists mainly of sulfate and carbonaceous aerosols trapped in the cold retracting polar dome in spring, coupled with reduced wet scavenging in winter and spring (Abbatt et al., 2019; Moschos et al., 2022).

Carbonaceous aerosols (here: total carbon, TC) consists of an organic fraction referred to as organic carbon (OC), and a refractory light-absorbing component named elemental carbon (EC) or equivalent black carbon (eBC) when quantified with

thermal-optical analysis or optical methods, respectively (Contini et al., 2018; Bond et al., 2013; Petzold et al., 2013). TC constitutes 20 to 90 % of the aerosol mass (Kanakidou et al., 2005; Putaud et al., 2010; Gentner et al., 2017). As a main PM component, it thus contributes to adverse effects on public health and climate. On the one hand, carbonaceous aerosols may contain toxic or carcinogenic compounds such as polycyclic aromatic hydrocarbons (PAH) (Mauderly and Chow, 2008; Kim et al., 2013; Smichowski et al., 2005; Daellenbach et al., 2020). On the other hand, both EC and OC are climate relevant:

The effective radiative forcing (ERF) for atmospheric aerosols is negative, and while the OC fraction has a negative ERF the EC fraction has a positive ERF (IPCC, 2021). Overall, the surface albedo for BC and OC on snow and ice is positive with a global mean ERF of 0.08 (0.00 to 0.18) (IPCC, 2021). Consequently, sources of OC, EC and subfractions must be understood to improve air quality and mitigate adverse effects of carbonaceous aerosols. Due to its complex composition and multitude of sources, however, carbonaceous aerosols are still inadequately understood.

Source apportionment is a widely used approach to gain understanding on emission, formation, and transformation of carbonaceous aerosols. It investigates the chemical and physical composition of aerosols at receptor sites to disentangle the contributions of individual emissions and the attribution to different source categories. Radiocarbon ($^{14}$C) measurements is an important source apportionment tool that can unambiguously separate between fossil and contemporary carbon present in carbonaceous aerosol, including in the OC and EC subfractions (Szidat et al., 2006; Winiger et al., 2015; Zotter et al., 2014).

Sources of OC and EC are often very different, and such additional information is obtained by means of $^{14}$C source apportionment of both EC and OC compared to a radiocarbon of TC analysis alone. The analysis of the OC subfractions water-soluble OC (WSOC) and water-insoluble OC (WINSOC) can lead to further information of the fossil and non-fossil fractions of the emitting sources (Zhang et al., 2014b).

Separation of OC and EC are method dependent, but the classification is widely recognised (Pöschl, 2003). EC is a primary

particle, i.e., emitted directly to the atmosphere, generated by incomplete combustion of fossil fuels and biomass, whereas OC is either primary or secondary, i.e., emitted directly or formed in the atmosphere by oxidation of both anthropogenic and biogenic precursor gases (Kanakidou et al., 2005). Thermal-optical analysis (TOA) is a well-established and commonly used technique for OC/EC determination (Chow et al., 2004; Cavalli et al., 2010; Chow et al., 1993; Schmid et al., 2001;





Huntzicker et al., 1982; Zenker et al., 2017; Dasari and Widory, 2022). Typically, two or more heating steps in an inert (i.e.,
helium) and in an oxidative atmosphere (i.e., 2 % oxygen in helium) are used to desorb OC and EC, respectively. During
analysis, the transmission or reflectance of the filter sample is continuously measured (Birch and Cary, 1996; Schmid et al.,
2001). A change in the transmission or reflectance signal indicates charring and EC loss. Charring is known as the process
when OC pyrolyses into EC, thus decreasing the transmission signal and creating a positive EC artefact (Cadle et al., 1980;
Yu et al., 2002). Charring leads to an overestimation of EC and an underestimation of OC. Additional to charring, some EC
is lost by desorption during thermal separation of OC, leading to a negative EC artefact. Both the positive EC artefact (i.e.,
charring) and the negative artefact (i.e., partial EC loss) may induce a bias to $^{14}$C measurement of EC. Charring adds OC,
which is typically more non-fossil than EC (Szidat et al., 2006, 2009; Zhang et al., 2012, 2014b; Zotter et al., 2014; Vlachou
et al., 2018), so that the measured $^{14}$C of EC may appear more non-fossil than it is. Partial EC loss usually affects non-fossil
EC (e.g., from biomass burning) more than fossil EC (e.g., from traffic or coal combustion) so that the remaining EC may be
altered and seem more fossil. A correction of both artefacts is therefore required for the accurate quantification of the fossil
vs. non-fossil shares of EC. EC recovery after OC/EC separation is determined using the transmission or reflectance signal
(Gundel et al., 1984; Zhang et al., 2012). Frequently used TOA protocols for OC/EC determination include EUSAAR_2
(Cavalli et al., 2010), IMPROVE (Chow et al., 1993), and NIOSH (Eller and Cassinelli, 1996). Radiocarbon measurement
requires a clear physical separation of OC and EC, since OC and EC do not originate from the same processes and often
show very different radiocarbon signatures (Szidat et al., 2006, 2007; Zhang et al., 2014b). Traditional TOA protocols may
still contain some OC in charred or an unaltered form after the split point, thus fail to perform the physical separation
adequately for radiocarbon source apportionment (Barrett et al., 2015; Zhang et al., 2012). Gustafsson et al. (2001)
developed a separation technique (CTO-375) in soil sediments, which was later applied to radiocarbon source apportionment
of atmospheric aerosols (Zencak et al., 2007). A two-step separation method developed by Szidat et al. (2004b) was utilised
for radiocarbon source apportionment (Zhang et al., 2010; Jenk et al., 2007; Szidat et al., 2004b). As these simplified
approaches still failed to provide an isolation of EC, our group (Zhang et al., 2012) established an improved four step
method (Swiss_4S) using water extraction before TOA and pure $O_2$ for an optimised EC recovery and reduced charring.
Later, Agrios et al. (2015) coupled the Sunset thermo-optical OC/EC analyser with on-line measurement in an accelerator
mass spectrometer (AMS) and implemented the previously developed Swiss_4S protocol.
Many have investigated EC in the Arctic including stable isotope ($^{13}$C) and radiocarbon analysis for source apportionment
(Winiger et al., 2016, 2017, 2015; Moschos et al., 2021). The fossil contribution of OC and WSOC is often not measured
directly but calculated by the isotope mass balance approach (Vlachou et al., 2018). Zhang et al. (2014a) lyophilised and re-
solubilised the eluate from water extraction before combustion in an elemental analyser coupled with radiocarbon
measurement. Menzel and Vaccaro (1964) as well as Sharp (1973) used potassium persulfate for the oxidation of dissolved
organic carbon in seawater. Lang et al. (2012) employed such a chemical wet oxidation for stable isotope analysis of
dissolved organic matter in freshwater samples. This method was later used for stable and radiocarbon analysis of marine




samples as well as compound-specific analysis of pyrogenic carbon (Lang et al., 2013; Wiedemeier et al., 2016), but has not been adapted for $^{14}$C analysis of WSOC from carbonaceous aerosols so far.

The present study provides a framework for an optimal OC/EC separation and radiocarbon analysis coupled with direct
$^{14}$C(WSOC) analysis (i.e., the $^{14}$C analysis of WSOC) by chemical wet oxidation applied on low-loaded Arctic filters. We provide a novel method for the EC yield extrapolation and charring correction based on a chemical desorption model that represent the behaviour of EC from different sources more realistically. Arctic filters were utilised as they are challenging for radiocarbon analysis due to their low loading and the large portion of pyrolysable species. Using an optimised strategy, we can measure the F$^{14}$C value (i.e., the Fraction Modern) in all major aerosol filter fractions (TC, EC, WSOC, WINSOC) with
the lowest possible amount of filter material.

## 2 Experimental

### 2.1 Overview of the analytical procedures

Aerosol filter samples were first water extracted to collect WSOC for subsequent radiocarbon measurement and to minimise formation of pyrolytic carbon (PC), caused primarily by WSOC, otherwise causing a dilution of the true EC signal. We then
used the first three steps of the Swiss_4S protocol (Zhang et al., 2012) to remove WINSOC from the filter by thermal-optical analysis, isolating EC. The filter's EC content were evolved by total combustion in a TOA analyser and subjected to on-line radiocarbon measurements. The WSOC eluate was converted to $CO_2$ by chemical wet oxidation before radiocarbon measurement. The following chapters explain the different procedures in brief, whereas the SI provides information that is more detailed.

### 2.2 Sampling and filter selection

Aerosol filter samples were collected between February 2017 and November 2018 at the Zeppelin Observatory (Svalbard) (78° 54′ N, 11° 52′ E) (475 m a.s.l.), which is part of the Global Atmospheric Watch (GAW) programme, the Arctic Monitoring and Assessment Programme (AMAP), and the European Evaluation and Monitoring Programme (EMEP) (Hung et al., 2010; Tørseth et al., 2012; Platt et al., 2022). Aerosol particles were collected on pre-fired (850 °C, 3 h) quartz fibre
filters (PALLFLEX® Tissuquartz 2500QAT-UP; 150 mm in diameter) downstream of a PM$_{10}$ inlet, using a Digitel high-volume sampler (DH-77, Hegenau, Switzerland). The sampler operated at a flow rate of 689 L min$^{-1}$, corresponding to an air volume of 6945 m$^3$ for a sampling time of one week. Filter samples were collected according to the quartz behind quartz (QBQ) set up (McDow and Huntzicker, 1990), allowing for an estimate of the positive sampling artifact of OC.

A fraction (46 mm diameter, corresponding to 16.6 cm$^2$) of the total filter area (153.9 cm$^2$) were cut for radiocarbon
measurement of $^{14}$C(TC), $^{14}$C(WSOC) and $^{14}$C(EC) (Fig. 1). The filter's TC, EC, and OC content were quantified according the EUSAAR_2 temperature programme (Cavalli et al., 2010), using transmission for charring correction. 18 filter samples





were received for radiocarbon measurement, but due to low EC loadings pooling of five subsequent filters was necessary (Fig. 1). Owing to the low filter loading, the water extraction for $^{14}$C(WSOC) and $^{14}$C(EC) was only performed on the front filters, whereas $^{14}$C(TC) analysis was performed on both front and back filters.

**2.3 Water extraction**

Three circular punches 22 mm (diameter) made from the 46 mm (diameter) aerosol filter were stacked and intercalated with silicone O-rings in 25 mm polycarbonate filter holders (Sartorius GmbH, Germany) with the exposed side facing upwards. A cleaned glass syringe (10 mL, ETERNA MATIC, Sanitex SA, Switzerland) was rinsed and filled with ultrapure water (18.2 MΩ·cm, Elga Purelab Flex 2, High Wycombe, UK) and attached to the filter holder with a 21G × 4 3⁄4 inch needle

(Sterican, B. Braun, Germany) at the filter holder outlet (Fig. 1). The needle pierced through a 12 mL EXETAINER® vial septum (12 mL, screw cap, item 938 W, Labco Ltd., Lampeter, UK). $5.0 \pm 0.2$ mL of water passed through the filters by gravity and collected in the EXETAINER® vials. Excess air could exit the vial by opening the screw cap half a turn before needle insertion. After water extraction, the vials were closed and stored at 4 °C until WSOC measurement. Excess water in the filter holder was removed using low-lint tissues and the water-extracted filters were dried overnight. The water-extracted

area (18 mm diameter) of the filter disc was punched out to remove the circumference that is not extracted, wrapped in aluminium foil, packed in air-tight plastic bags, and stored in a freezer at -20 °C for subsequent WINSOC removal.

**2.4 WINSOC removal**

WINSOC was removed from the water-extracted filters using a thermal-optical OC/EC analyser (Model 5L, Sunset Laboratories Inc., USA) for separation of EC. WINSOC removal was performed with the first three steps of the Swiss_4S

protocol, thus denoted as Swiss_3S. This allows for individual WINSOC removal runs and pooling of several filters for $^{14}$C(EC) analysis. The water-extracted filters were cut in quadrants (0.64 cm² each) to fit the OC/EC analyser sample holder (10 × 15 mm). Up to 12 WINSOC removal runs per single sample and 24 runs for pooled samples were performed. After WINSOC removal, the filters were stored in a freezer (−20 °C) until $^{14}$C(EC) analysis. In the final step, EC was combusted in the thermal-optical OC/EC analyser subjected to online radiocarbon measurement (Agrios et al., 2015). The protocol was

modified to compensate for EC losses (see section 2.10) observed with the standard protocol (Zhang et al., 2012). WINSOC removal was performed in these three steps: step 1 (pure $O_2$, 375 °C, 240 s), step 2 (pure $O_2$, 425 °C, 120 s), and step 3 (pure He, 600 °C, 120s). This procedure provided EC yields >0.7.

**2.5 Direct $^{14}$C(WSOC) measurement**

Inorganic carbonaceous impurities were removed by acidification and helium flushing. For this, $H_3PO_4$ (0.5 mL 8.5 %)

freshly prepared from $H_3PO_4$ (85 %, Suprapur grade, Merck KGaA, Germany) was added using a 1 mL Hamilton (Reno, NV, USA) glass syringe, and high-purity (99.999 %) helium was purged (50 mL min$^{-1}$) through the sample at room temperature for 3 min. The sample septum was pierced with a custom-made needle with a gas inlet and outlet hole, where





the gas outlet was submerged (~1 cm) and the gas inlet was placed in the upper part of the headspace. These steps were robotically performed by a PAL HTC–xt (CTC Analytics AG, Switzerland) mounted on top of a carbonate handling system (CHS, Ionplus AG, Switzerland).

The chemical wet oxidation procedure was used to oxidise WSOC to $CO_2$ for radiocarbon measurement (Lang et al., 2012; Wiedemeier et al., 2016). The oxidiser (10 % potassium persulfate (ACS grade, Sigma-Aldrich, USA)) was freshly prepared, dissolved in $H_3PO_4$ (5 %, m m$^{-1}$), pre-oxidised (90 °C, 30 min), and flushed with helium (50 mL min$^{-1}$, 3 min) to remove all carbonaceous contaminants. Oxidiser (0.25 mL) was added to each sample and the reaction progressed overnight at 75 °C on the hot plate of the CHS. For sampling the generated $CO_2$ (50 mL min$^{-1}$, 3 min), we used the custom-made needle and PAL autosampler described above. The CHS was connected to a custom-built water trap to retain liquid water in a wash bottle (25 mL), whereas the remaining water vapour was trapped using $P_2O_5$ (SICAPENT®, Merck KGaA, Germany). The dry gas was then carried to the gas interface system (GIS) and trapped on a X13–zeolite trap (Ruff et al., 2007; Wacker et al., 2013). After sampling, the trapped $CO_2$ was thermally released and mixed with helium for $^{14}C$ measurement. We applied a cross-contamination of 0.5 % and a constant contamination of $0.9 \pm 0.2$ µg C with $F^{14}C = 0.20 \pm 0.08$ on samples subjected to chemical wet oxidation (see Text S5).

## 2.6 Online $^{14}C$(TC) and $^{14}C$(EC) measurement

5.2 cm$^2$ of each filter (16.6 cm$^2$) was used for $^{14}C$(TC) analysis and 10.4 cm$^2$ for pooled samples. $^{14}C$(TC) was measured by complete combustion (240 s, 870 °C, pure $O_2$) in the Sunset OC/EC analyser before $^{14}C$ analysis (see section 2.7). Complete combustion was ensured by passing through the second furnace of the analyser containing $MnO_2$ at 870 °C. The evolved $CO_2$ was analysed by the non-dispersive infrared (NDIR) detector, resulting in 20.2–116.2 µg C and 27.0–99.3 µg C for single and pooled filters, respectively. An equivalent area was used for back filters, yielding 3.4–11.3 µg C and 6.2–11.8 µg C for single and pooled filters, respectively.

For $^{14}C$(EC) analysis, the filters consisting of only EC after water extraction (see section 2.3) and WINSOC removal (see section 2.4) were combusted in the Sunset OC/EC analyser. Between 3.8 to 15.3 cm$^2$ of filter material was combusted for EC, yielding 3.9–16.8 µg C. After combustion, the released gas was dried ($P_2O_5$, SICAPENT®, Merck KGaA, Germany) and transferred to the GIS where $CO_2$ was trapped and thermally released for on-line measurement in the AMS (Agrios et al., 2015) (see section 2.7). We applied a cross-contamination correction of 0.2 % due to a $CO_2$ adsorption memory effect on the zeolite trap for TC and EC (Salazar et al., 2015). A constant contamination correction of $0.40 \pm 0.20$ µg with $F^{14}C = 0.80 \pm 0.36$ was applied. To account for EC loss and charring during TOA, $F^{14}C$(EC) values were corrected using the "COMPYCALC" script (see section 2.10).

## 2.7 Radiocarbon measurement

Radiocarbon measurement was performed using a MICADAS (Mini radioCArbon DAting System) accelerator mass spectrometer (AMS) at the University of Bern (Synal et al., 2007; Szidat et al., 2014; Fahrni et al., 2013). On each AMS



measurement day, multiple OxII (Oxalic Acid II, SRM 4990 C, National Institute of Standards and Technology, NIST, Gaithersburg, USA) and fossil NaAc (sodium acetate, Sigma-Aldrich, No. 71180) (Szidat et al., 2014) standards were analysed. BATS software version 3.6 (Wacker et al., 2010) was used for standard normalisation as well as data correction for background, blank, and mass-fractionation.

**2.8 Contamination precautions**

All filter handling and water extraction was performed in a laminar flow cabinet. All glassware was cleaned using $H_3PO_4$ (1M, ACS grade, Merck KGaA, Germany) and pre-fired (500 °C, 5 h), as described by Lang et al. (2012). The vials were leak tested overnight at 75 °C and ~4 bar of $N_2$. The glass syringe used for water extraction was rinsed before use using ultrapure water and then pre-fired (500 °C, 2 h). The filter holders and silicone O-rings were rinsed and sonicated with ultrapure water before use and dried in a laminar flow cabinet.

**2.9 EC correction model**

OC/EC separation leads to losses of EC during thermal desorption, which needs to be corrected by an $F^{14}C(EC)$ yield extrapolation. The correction supposes that the EC fraction consists of two subfractions, a subfraction with certain volatility at the temperature of steps S1, S2 and S3 and a refractory subfraction. The yield ($Y$) and $F^{14}C$ of EC ($F_{EC}$) of the mixture are empirically determined as explained in sections 2.10 and 2.6, respectively. For further information, $Y$ and $F_{EC}$ are modelled 205 from the mass balance as follows:

$$Y = \frac{m_v + m_{nv}}{m_{v0} + m_{nv0}} = \frac{q_m * \alpha_v + \alpha_{nv}}{q_m + 1} \tag{1}$$

$$F_{EC} = \frac{m_v * F_v + m_{nv} * F_{nv}}{m_v + m_{nv}} = \frac{q_m * \alpha_v * F_v + \alpha_{nv} * F_{nv}}{q_m * \alpha_v + \alpha_{nv}} \tag{2}$$

$$q_m = \frac{m_{v0}}{m_{nv0}} \tag{3}$$

The parameter $q_m$ is the quotient of the initial masses of the non-refractory ($m_{v0}$) to refractory ($m_{nv0}$) subfractions and it is 210 calculated with Eq. 3. $F_v$ and $F_{nv}$ are the Fraction Modern of the non-refractory ($F^{14}C = 1$) and refractory ($F^{14}C = 0$) subfractions. $\alpha_v$ is the mass fraction of the non-refractory EC subfraction that withstands the WINSOC removal procedure relative to the initial mass calculated as $\alpha_v = m_v m_{v0}^{-1}$. $\alpha_{nv}$ is the analogue of $\alpha_v$ for the refractory subfraction. Each step of the WINSOC removal has a value of $\alpha$, which is calculated with Eq. 4 by a first-order kinetic equation

$$\alpha = e^{-t*K(T)} = e^{-t*K(T_{ref})e^{\left(\frac{E_a}{RT_{ref}} - \frac{b*E_a}{RT}\right)}} \tag{4}$$

where $t$ is the step desorption time (s) and the desorption rate $K$ (s$^{-1}$) is calculated with the temperature-dependent Arrhenius equation. The global $\alpha$ is the joint yield of all the steps $\alpha = \alpha_1 * \alpha_2 * \alpha_3$. Bedjanian et al. (2010) also used a first-order kinetic coupled to Arrhenius for investigating the thermal desorption of polyaromatic hydrocarbons (PAH) from soot surfaces. The main composition of EC fraction is soot with compounds molecularly similar to PAHs of diverse sizes. Bedjanian et al. (2010) found that the activation energy ($E_a$) for PAH is in the range of 85 kJ mol$^{-1}$ to 134 kJ mol$^{-1}$ linearly depending on the



molecular weight for the range of 178-302 g mol$^{-1}$. The desorption rate $K$ was ranging from $3 \times 10^{-3}$ s$^{-1}$ to $5 \times 10^{-5}$ s$^{-1}$ for a
temperature range of 370–350 K. The Arrhenius pre-exponential factor was solved by using the concept of the reference
temperature (Peleg et al., 2012; Schwaab and Pinto, 2007). The scale of the desorption rate $K$ is logarithmic, meaning that a
small increase or decrease in temperature leads to a substantial change in the desorption rate. Our optimised $E_a$ is
100 kJ mol$^{-1}$, and our reference desorption rate $K$ is $1.5 \times 10^{-6}$ s$^{-1}$ at 340 K ($T_{ref}$) which is in the range of the desorption rates

from Ghosh et al. (2001) converted from room temperature to our reference temperature. The data can be found in Table 3 of
Ghosh et al. (2001) with values between $1.2 \times 10^{-9}$ to $3.6 \times 10^{-9}$ s$^{-1}$ at 293 K ($E_a = 116$ to 133 kJ mol$^{-1}$), which results in
desorption rates at $T_{ref} = 340$ K of $9 \times 10^{-7}$ to $7 \times 10^{-6}$ s$^{-1}$. The activation energy for the refractory fraction is unknown, but
we may assume that the molecular weights of the compounds of the refractory fraction are much heavier. Bedjanian et al.
(2010) showed a linear relationship between molecular size and volatility with $E_a$; therefore, we introduce an empirical

factor $b$, which represents how much bigger $E_a$ is for the refractory relative to the non-refractory fraction as shown in Eq. 5.
$E_a$ and $K(T_{ref})$ values were kept within the references ranges and optimised with the data from our previous works (see
section 3.1 and Fig. S2 in Zotter et al., 2014); $E_a$ and $K(T_{ref})$ were taken from the references; $t$ and $T$ were fixed to the
WINSOC removal conditions.

$$E_{a_{nv}} = b * E_{a_v} \tag{5}$$

The values for the parameters $b$ and $q_m$ are optimised for each individual sample as follows. The $q_m$ and $b$ parameters are
selected, the mathematical model estimates $\alpha$ for both refractory and non-refractory fractions with Eq. 4 and Eq. 5. Then the
yield and $F_{EC}$ are calculated with Eq. 1 and Eq. 2. The yield and $F_{EC}$ from the model are compared with the empirical yield
and $F_{EC}$ using a cost function shown in Eq. 6. The cost function is minimised by a gradient descent method from the R script.
$q_m$ and $b$ are not general parameters or general coefficients; usually their values are different between samples because their

molecular compositions are different. The number of data values in the cost function is only two.

$$J(q_m, b) = \left[F_{EC,data} - F_{EC,model}(q_m, b)\right]^2 + \left[Y_{data} - Y_{model}(q_m, b)\right]^2 \tag{6}$$

Our model is a two-component model used to describe a multicomponent system. Two-component models are common: for
example, the Keeling approach to describe the mixing of one component onto a background component in complex
atmospheric air or dissolved organic carbon in ocean waters (Keeling, 1958; Walker et al., 2016). Each refractory and non-

refractory subfraction are composed of a complex mixture of compounds with a continuum of volatilities and $^{14}$C content.
However, the mean desorption energy of the subfractions obeys Eq. 5. The $^{14}$C content of both subfractions is not exactly 1.0
or 0.0 but a continuum where the mean F$^{14}$C of the refractory subfraction trends to fossil values while the opposite occurs to
the non-refractory subfraction.

## 2.10 EC and OC correction calculations

The F$^{14}$C(EC) yield extrapolation and charring correction was performed with a script named COMPYCALC
(COMprehensive Yield CALCulation) written in R (R Core Team, 2020), available on GitHub (github.com/martin-



rauber/compycalc) and archived in Zenodo (Rauber and Salazar, 2022). Using Eq. 7, an initial value of $F^{14}C(OC)$ is calculated prior running the script using the uncorrected $F^{14}C(EC)$ value, as $F^{14}C(OC)$ is needed for the charring correction (see Table S1). $F_{TC}$ and $F_{EC}$ are the radiocarbon values (Fraction Modern, $F^{14}C$) for TC and EC before correction, respectively, whereas $r$ is the EC/TC ratio.

$$F_{OC} = \frac{F_{TC} - F_{EC}*r}{1-r} \tag{7}$$

The EC yield was calculated using the laser transmission signal (655–660 nm) of the OC/EC analyser. Each WINSOC raw data file from the Sunset OC/EC analyser is loaded by the COMPYCALC script. The laser transmission is dependent on the temperature (Peterson and Richards, 2002). By applying a correction on the complete laser signal of the thermogram, this temperature–induced change in transmission is accounted for. For COMPYCALC, a generic file corresponding to the S4 step in the Swiss_4S protocol is used for the calculation of the temperature dependence correction of the laser transmission signal. The EC yield ($Y$) after the three WINSOC removal steps was calculated as the ratio of the attenuation (ATN) after S3 to the initial ATN after water extraction. ATN is a unitless parameter proportional to the light-absorbing EC mass calculated using the Beer-Lambert Law and the laser transmission signal (Gundel et al., 1984; Zhang et al., 2012). Here, the temperature-dependence correction of the laser transmission signal is applied. Formation of pyrolysed OC (i.e., charring, see below) is quantified by the ratio of the difference between the maximum ATN and the initial ATN of each step (Gundel et al., 1984; Zhang et al., 2012; Vlachou et al., 2018). When filter punches do not cover the sample holder spoon area completely, small filter movements from vibrations caused by the OC/EC analyser may occur. This may inflict faulty laser signals when filters are smaller than the sample holder area (10 × 15 mm). WINSOC removal is usually performed on multiple filter cuts and EC yield and charring is calculated for each filter cut. COMPYCALC filters by the interquartile range of < 1.5 individually for EC yield and charring in S1, S2, and S3, and removes the row(s) containing outliers in the data frame. The number of filters cuts used for calculation is summarised in Table S5. The COMPYCALC summary output (see Fig. S2 and Table S2) only includes the filtered data, however, the raw data (not filtered) is preserved and given as an output as well. The EC yield and charring before filtering is shown in Table S6.

The measured $F^{14}C(EC)$ values ($F_{EC}$) were extrapolated to 100 % EC yield ($F_{EC(corr)}$) using Eq. 9 to account for the EC loss during WINSOC removal. For the empirical data, the yield $Y$ and the $F_{EC}$ are directly measured while $\alpha$ is calculated with Eq. 4 The reader must note that Eq. 8 is obtained when Eq. 1 is input in the denominator of Eq. 2 and solving for parameter $q_m$. If $Y = 1$, then Eq. 8 becomes the $F_{EC}$ extrapolated at 100 % yield (Eq. 9).

$$F_{EC} = \frac{q_m*\alpha_v*F_v + \alpha_{nv}*F_{nv}}{Y(1+q_m)} \tag{8}$$

$$F_{EC(corr)} = \frac{q_m*F_v+F_{nv}}{1+q_m} \tag{9}$$

Beside extrapolation to 100 % EC yield, the Fraction Modern must be corrected for charring as some OC is pyrolysed into EC. The charring corrected Fraction Modern ($F_{charrA}$) is calculated in Eq. 10 using the Fraction Modern of EC ($F_{EC(corr)}$) extrapolated to 100 %. Fraction Modern of OC ($F_{OC}$) was previously calculated using Eq. 7, $\varepsilon$ is the total charring. It is





assumed that 50 % of the pyrolysed OC is lost in the subsequent temperature steps as EC loss again, thus a factor of 0.5 is

used for correction of these losses of pyrolysed OC (Zotter et al., 2014). For Eq. 11, the Fraction Modern of EC without extrapolation to 100 % EC yield is used. In Eq. 12, the Fraction Modern with charring correction ($F_{charrC}$) is calculated with the charring correction slope $\beta$ and EC yield ($Y$). $\beta$ is the slope between the Fraction Modern and EC yield as defined previously (Zotter et al., 2014; Zhang et al., 2012). The final Fraction Modern with charring correction in Eq. 13 is calculated as the mean of Eq. 10 and Eq. 12.

$$F_{charrA} = \frac{F_{EC(corr)} - F_{OC} * 0.5 * \varepsilon}{1 - 0.5 * \varepsilon} \tag{10}$$

$$F_{charrB} = \frac{F_{EC} - F_{OC} * 0.5 * \varepsilon}{1 - 0.5 * \varepsilon} \tag{11}$$

$$F_{charrC} = \beta * (1 - Y) + F_{charrB} \tag{12}$$

$$F_{EC(final)} = \frac{F_{charrA} + F_{charrC}}{2} \tag{13}$$

After all calculations, a data file with overall EC yield, the charring contribution for each OC removal step (S1, S2, S3), the

total charring contribution as well as the $F^{14}C$(EC) input value $F_{EC}$, $F^{14}C$(EC) extrapolated to 100 % EC yield ($F_{EC(corr)}$), and $F^{14}C$(EC) extrapolated to 100 % EC yield and corrected for charring ($F_{EC(final)}$) is generated as an output. The final $F^{14}C$(OC) is calculated using Eq. 7 with $F_{EC(corr)}$ and reported as $F_{OC(corr)}$.

**2.11 EC yield calculation and WINSOC amount calculation**

EC yield calculation and amount calculation of each WINSOC step was performed with the R script "Sunset-calc", written

as an R Shiny application (R Core Team, 2020; Chang et al., 2017). Sunset-calc provides amount calculation for each step in the Swiss_3S and Swiss_4S protocols (Zhang et al., 2012) as well as EC yield and charring calculation (see Table S7). Furthermore, EC yield and charring corrected OC (WINSOC) and EC amounts are calculated (see Table S4). The Sunset OC/EC analyser raw files are loaded in a web graphical user interface and the results are received as a downloadable file. EC yield and charring calculation is based on COMPYCALC as described in 2.9. The amount calculation is made with an

integration of the NDIR signal. The application has been deployed on an R server (14c.unibe.ch/sunsetcalc). Sunset-calc is available on GitHub (github.com/martin-rauber/sunset-calc) and archived in Zenodo (Rauber, 2021).

**3 Results and Discussion**

**3.1 Validation of the correction**

Figure 2a shows the comparison of the modelled $F_{EC}$ versus the empirical $F_{EC}$, and Fig. 2b shows the modelled EC yield

versus the empirical EC yield. The empirical data is taken from Fig. S2 of our previous work (Zotter et al., 2014). Figures 2a and 2b indicate that our model provides good accuracy for predicting the FEC and the EC yields. We determined a relative accuracy of 109 ± 4 % as an agreement of the measured values compared to the modelled values using a linear model and its residual standard uncertainty. Therefore, the $b$ and $q_m$ values are reliable. Figure 2c indicates that the $b$ parameter falls into



two volatility groups. The group close to $b = 1.0$ and the group mainly within 2.0 to 2.5. These are interesting results as the initial value for $b$ is 2.0 at the start of the gradient descend optimisation. We examined the optimisation again and the script does check values in the range of 1.0 to 2.0. Figure 2c is an indirect probing of the volatility of the sample compounds. Figure 2d shows the calculated parameters for each sample revealing that $q_m$ increases with $F_{EC}$. This indicates that for higher $F_{EC}$ values, closer to the atmospheric non-fossil levels, the initial mass of the non-refractory biogenic EC (section 2.9) subfraction must be higher than the initial mass of the more fossil refractory EC subfraction.

Figure 2e provides examples of the modelling of the $F_{EC}$ versus the modelled EC yields for different values of the parameter $b$. The EC yield is decreased by proportionally increasing the temperature of each of the three steps of the WINSOC removal. The model allows us to extrapolate the $F_{EC}$ value of any sample with a yield lower than 100 % to the $F_{EC}$ value corresponding to 100 % yield, which defines the correction for EC loss. According to the Arrhenius approach, the model has a non-linear shape which may be approximated by a linear model in the region of EC yields higher than 0.5. Before developing this non-linear model, we applied a simple linear model for the EC loss correction according to previous publications (Zotter et al., 2014). The measurement conditions usually keep the EC yield higher than 0.4, thus the linear model remains useful under certain conditions. Nevertheless, the non-linear model is superior and shall be used in future. Figure 2f is similar to Fig. 2e but for different $q_m$ values. As shown in Zotter et al. (2014), different samples may show different slopes and intercepts for the linear model. Figure 2e and Fig. 2f show that different values of $b$ and $q_m$ explain the different slopes and intercepts observed previously in the data. Extrapolation and correction to $F_{EC(corr)}$ of the data from Zotter et al. (2014) is shown in Fig. S6. In Fig. S6, same-colour results belong to punches from the same filter, however the experimental conditions of their online TC/EC measurements were variated in order to obtain different yields and $F_{EC}$ values. Therefore, the same-colour results in Fig. S6, ideally, should have the same $F_{EC}$ value extrapolated to 100 % yield. As indicated in section 2.9, this data was useful to optimise the $E_a$ and $K(T_{ref})$ values by minimising the differences between the yield-corrected $F_{EC}$ of the same-colour results. This optimisation was performed prior to the application of the non-linear model to the results of this paper.

## 3.2 Concentrations of carbonaceous aerosols

Results from the 21-month sampling period (Table 1) showed a mean TC concentration of 137 ng C m$^{-3}$ (range: 65–264 ng C m$^{-3}$) and a mean EC concentration of 14 ng C m$^{-3}$ (range:3–40 ng C m$^{-3}$), resulting in a mean OC/EC ratio of 11.7 (range: 4.5–27). The filter sampled from 28 September to 06 October 2017, had elevated TC (601 ng m$^{-3}$) and EC (52 ng C m$^{-3}$) levels, and were excluded from the mean reported above as this would clearly distort the mean. The OC/EC ratio for this filter sample was 10.5 and thus comparable to the mean of the other samples. For 5 of the 13 samples, two consecutive filter samples were pooled to obtain a sufficient carbon amount for $^{14}$C analysis (see Table 1). Lower TC values were seen in winter (November to March) compared to summer (April to October), whereas it was the other way around for EC. Consequently, the OC/EC ratio shows a seasonality with lower values in winter and higher in summer. TC on back filters had a mean concentration of 152 ng C m$^{-3}$ (range: 63–254 ng C m$^{-3}$) and showed no seasonality. The mean pure





WINSOC concentration (Table 2), corresponding to Step 1 of the Swiss_3S protocol, was 26 ng C m$^{-3}$ (range: 9–71 ng C m$^{-3}$), whereas the mixed (WINSOC + EC) S2 and S3 fractions had mean concentrations of 4 ng C m$^{-3}$ (range: 0.5–26 ng C m$^{-3}$) and 7 ng C m$^{-3}$ (range:1.5–16 ng C m$^{-3}$). The aforementioned high loading filter sample from the transition

September/October 2017 (111 ng C m$^{-3}$ (S1), 26 ng C m$^{-3}$ (S2), and 27 ng C m$^{-3}$ (S3)) were excluded from the mean. The total amount of WINSOC including EC loss was 37 ng C m$^{-3}$ (range:1.5–16 ng C m$^{-3}$, excluded filter: 164 ng C m$^{-3}$). WSOC was calculated by subtracting EC and total WINSOC from TC, which gave a mean of 39 ng C m$^{-3}$ (range: 0.5–92 ng C m$^{-3}$). The September/October 2017 filter sample had a loading of 284 ng C m$^{-3}$and was excluded from the mean. The charring and EC loss corrected mean amount calculated with Sunset-calc (see section 2.11, Table S4) for WINSOC was

34 ng C m$^{-3}$ (range: 11–90 ng C m$^{-3}$, excluded filter: 151 ng C m$^{-3}$) and the mean corrected amount for EC was 15 ng C m$^{-3}$ (range: 3.7–39 ng C m$^{-3}$, excluded filter: 67 ng C m$^{-3}$). For these calculations and corrections, the R Shiny application Sunset-calc was necessary as this is not possible with the default software tools provided for the Sunset OC/EC analyser. $^{14}$C(TC) measurements on back filters (see Table 3) revealed a mean filter loading of 90 ng C m$^{-3}$ (range: 26–189 ng C m$^{-3}$) excluding the autumn 2017 filter, which had a back filter loading of 501 ng C m$^{-3}$.

### 3.3 Development of preparation methods

#### 3.3.1 Water extraction

For water extraction, three filter punches were stacked to maximise the amount of extractable WSOC. Prior to filter sample extraction, trials with empty filters and the screw type polycarbonate water extraction unit were made. Stacking more than three filters was not feasible, as it makes the water extraction housing prone to leakage. The sample water extraction was

gravity-fed. Ultrapure water was filled in the pre-combusted glass syringe directly from the tap of the ultrapure water system and screwed onto the previously assembled water extraction unit to avoid unnecessary liquid transfer. The extraction of 5 mL took 2-3 min depending on the number of filters stacked.

The water-extracted filter material was subjected to WINSOC removal and EC measurement. Elimination of WSOC is beneficial as it is shown to pyrolyse into EC (charring) when subjected to thermal-optical analysis (Yu et al., 2002; Cadle et

al., 1980). The F$^{14}$C(OC) is generally higher than for F$^{14}$C(EC) (Szidat et al., 2004b, 2009; Zhang et al., 2012), but often exceeded by F$^{14}$C(WSOC) due to substantial contributions from biogenic sources and biomass-burning emissions (Zhang et al., 2014a; Kirillova et al., 2013; Weber et al., 2007). Therefore, a small contribution of charred OC significantly biases the measured F$^{14}$C of the EC fraction, which is prevented by the WSOC removal.

#### 3.3.2 Adaptations of the OC/EC analyser for WINSOC removal

The filter holders for water extraction are of screw type, thus round punches were required for water extraction. For WINSOC removal, a single layer of filter material cannot exceed the area (1.5 cm$^2$) of the sample holder spoon in the Sunset OC/EC analyser. Although it is not necessary to fully cover the sample holder area, the filter cut should cover most the area



to utilise the laser transmission signal for calculations. Stacking of filters should be avoided, as lower filters may not encounter the same conditions as the topmost filter, especially in terms of oxygen supply, which may cause differences with respect to both charring and EC losses within the stack. Furthermore, calculating an EC yield is not feasible after stacking two or more filters. We observed spikes in the laser transmission signal for small filter punches (<0.5 cm$^2$), possibly due to filter movements caused by instrument vibrations. Due to the limitation of circular cuts for water extraction and a rectangular shaped sample holder in the OC/EC analyser, the water-extracted filter was cut in quadrants. This enables the complete use of filter material; however, at the expense of a more labour intensive WINSOC removal. The three water-extracted punches from each filter were cut into 12 quadrants and 24 for each pooled sample. WINSOC was then removed from each sector using the Swiss_3S protocol (Zhang et al., 2012), requiring 18.5 min per run. High EC losses were observed with the standard Swiss_3S protocol, hence the protocol was adapted. Decreasing the temperature from 450 to 425 °C in S2 and from 650 to 600 °C in S3 increased EC yields from < 0.4 to 0.6. Shortening the 600 °C pure He step in S3 from 180 s to 120 s, further reduced EC losses, leading to a mean EC yield of 0.87 (range: 0.72–0.95) (Figs. 3 and 4). As shown in Fig. 4, the average charring after WINSOC removal was 2.8 % (range of 1–6.8 %) for S1, 0.6 % (0–2.4 %) for S2, and 3 % (1.3–9.0 %) for S3, with a total charring of 6.5 % (2.5–12.9 %). The OC and EC concentrations must be corrected for charring and EC losses using Sunset-calc (see sections 2.11 and 3.2). This enables a simple WINSOC removal protocol optimisation and adaptation after each run. The outcome of Sunset-calc is also employed for the correction of biases of $^{14}$C(EC) results caused by charring and EC losses (see section 3.4.1).

In the present work, WINSOC was removed, but not subjected to radiocarbon measurement due to the very low filter loading. In the Swiss_3S protocol, only the S1 fraction consists of pure WINSOC, as S2 and S3 are considered a mixture of WINSOC and EC. The average WINSOC loading in S1 was 1.8 µg C cm$^{-2}$, ranging from 0.9 to 3.7 µg C cm$^{-2}$, whereas radiocarbon measurements require at least 3 µg C. With higher loaded filters, $^{14}$C(WINSOC) measurements can be implemented in the workflow presented.

### 3.3.3 Wet oxidation and WSOC measurement

Filter extraction and chemical wet oxidation may add contaminants and stringent preparations (section 2.5) were needed to ensure low procedural blanks. This included the use of acid-cleaned (high purity grade H$_3$PO$_4$) and baked out glassware, and pre-oxidation of the oxidiser solution used to remove contaminants. The freshly prepared oxidiser solution was pre-oxidised at 90 °C for 30 min before helium flushing with helium to remove carbonaceous contaminants. This step removes contaminants in the oxidiser itself as well as in the ultrapure water and equipment used. The oxidiser concentration was increased to 10 % from 4 %, whereas the amount of oxidiser added to the sample was reduced to 0.25 mL from 1 mL, compared to Lang et al. (2012). Oxidation was performed at 75 °C overnight, deviating from previous studies by Lang et al. (2012) (100 °C for 60 min) and Lang et al. (2013) (90 °C for 30 min). EXETAINER® vials store gas with little leakage even after multiple needle punctures (Glatzel and Well, 2008). Al vials used for samples, standards and blanks were leak tested before use (section 2.8) at the same temperature (75 °C) as the oxidation step takes place. Vials are more prone to leakage at



higher temperatures; hence we lowered the reaction temperature to 75 °C. Both leak testing and a lower reaction temperature kept loss of precious sample material at a minimum. The sample acidification, helium flushing, and chemical wet oxidation was performed the day before measurement. The butyl rubber septum of the EXETAINER® may contaminate the sample over time when exposed to the strongly acidic and oxidative environment. As a cautionary principle, samples should be

measured the day after preparation to minimise any losses, contaminations, and potential isotopic fractionation. In the present work, helium was purged at 75 °C with the gas needle through the oxidised sample, unlike Lang et al. (2012), where only the headspace was sampled at room temperature. Considerable amounts of liquid (~0.3 mL per sample) that were carried with the gas were trapped in a custom-build gas wash bottle (25 mL). Remaining water vapour was removed by a Sicapent® trap ($P_2O_5$ on inert carrier material) to protect the zeolite trap in the gas interface system (GIS). The $CO_2$ amount

was determined by the GIS pressure gauge based on the ideal gas law before dilution with helium and feeding the gas mixture into the ion source of the AMS. This procedure provides an estimation of the amount of WSOC only.

### 3.3.4 Procedural blank

The WSOC procedural blank was determined by performing the water extraction and wet oxidation procedure, using pre-baked (2 h, 750 °C) quartz fibre filters (PALLFLEX® Tissuquartz 2500QAT-UP), as described in section 2.3. After

extraction, different amounts of OxII (SRM 4990 C) or fossil NaAc solutions (~1000 ppm) were added to the vials and subjected to chemical wet oxidation (section 2.4). The mass and Fraction Modern of the contaminant was determined based on the constant contamination approach by a drift model (Hanke et al., 2017; Salazar et al., 2015) (see Supplementary Material, Fig. S7). In previous studies, the WSOC eluate was dehydrated by lyophilisation before re-dissolving and combustion in an elemental analyser coupled to an AMS (Zhang et al., 2014a). Compared to the lyophilisation method, the

procedural blank was lower for chemical wet oxidation, with a mass of contamination of $0.9 \pm 0.2$ µg C and the corresponding $F^{14}C$ of $0.20 \pm 0.08$.

### 3.4 Radiocarbon results

### 3.4.1 Correction of the $^{14}C(EC)$ results

Early approaches of $^{14}C(EC)$ measurements focused on the separation of OC and EC (Zhang et al., 2012; Barrett et al., 2015;

Zencak et al., 2007), however, some OC pyrolyses into EC creating a positive artefact, and some EC is lost by desorption, degradation or oxidation (Cadle et al., 1980; Yu et al., 2002; Gundel et al., 1984; Zhang et al., 2012), but efforts to correct $^{14}C(EC)$ were not considered then (Szidat et al., 2006, 2004b, a; Dusek et al., 2014; Andersson et al., 2011; Bernardoni et al., 2013). Zhang et al. (2012) implemented a linear correction for EC losses to account for the underestimation of biomass burning EC. The composition of OC and EC underlies spatial and temporal variability and thus the linear correction slope

will differ. Zotter et al. (2014) addressed this issue by introducing different slopes for winter and summer, as the linear correction slope for EC differs considerably between these two seasons. Consequently, the linear correction slope must either





be established for each site with multiple EC yield measurements or estimated based on previous measurements. For low-loaded filters and for sites with limited filter availability such as the Arctic, this can be a particular challenge. Here, we apply an optimised approach, using COMPYCALC that combines the determination of both EC losses and EC bias from charring

of OC with the thermal desorption model (section 2.10). Furthermore, COMPYCALC uses the basis of Zhang et al. (2012) for the EC yield calculation and the charring calculation, where the attenuation (ATN, section 2.10) calculated from the laser transmission signal is used. Charring correction after EC yield extrapolation was performed in accordance with Zotter et al. (2014), assuming that half of the pyrolytic EC that forms during the analysis is lost by the last heating step during WINSOC removal. Table 4 summarises EC and OC before and after corrections for EC yield and charring. The initial $F^{14}C(OC)$ value

($F_{OC}$) is calculated with the initial EC value ($F_{EC}$) for correction. As described in section 2.10, the COMPYCALC script is run for the extrapolation of EC yield and charring correction to yield the final corrected EC value ($F_{EC(final)}$). Then, using $F_{EC(final)}$, the final OC value ($F_{OC(final)}$) is calculated.

### 3.4.2 Quality aspects of the $F^{14}C(OC)$ calculation

Thermal-optical OC/EC separation discussed in the present work focuses on EC and WSOC and the optimisation thereof.

Early work on $^{14}C$ analysis did not include measures to reduce charring, which included substantial biases in the $^{14}C$ analysis particularly for EC but also for OC, as $^{14}C(OC)$ was determined directly by combustion of the filters in oxygen at 340 °C (Szidat et al., 2004b). Later work included water extraction for charring reduction of EC (Yu et al., 2002; Novakov and Corrigan, 1995). Zhang et al. (2012) combined water extraction with an optimised four-step protocol and, thus, further improved OC/EC separation. However, only S1 was considered as pure OC in this first TOA protocol and thus may include

two possible biases of the $^{14}C(OC)$ result, as different OC fractions were not considered: first, the portion of OC that undergoes charring in S1 and, thus, is shifted to later steps, and second, more refractory OC that evolves during S2 and S3. This flaw was improved later by Zhang et al. (2015) by omitting the direct $^{14}C$ measurement of OC, calculating $F^{14}C(OC)$ as the difference between $F^{14}C(TC)$ and $F^{14}C(EC)$, as it is in the present study (Eq. 7). Hence, a better OC/EC separation improves both the quality of the measured $F^{14}C(EC)$ value and the calculated $F^{14}C(OC)$ value.

### 3.4.3 Measurement limitations

Radiocarbon measurement requires a minimum of 2-3 µg C per sample disregarding of the hyphenation method (Wacker et al., 2013). With the setup used in the present work, the water extraction method is limited by extraction setup diameter and the number of punches to be stacked. Accordingly, for WSOC a minimum filter loading of 0.3 µg C cm$^{-2}$ is required. Within reason, there is no known limit for the chemical wet oxidation. Radiocarbon measurements coupled with the Sunset OC/EC

analyser are limited by the sample holder, allowing for stacking up to six rectangular 1.5 cm$^2$ filters punches (9 cm$^2$ in total). In the present work, the remains after punching out the circular filters for WSOC were used for TC, which makes it difficult to fit the material on the regular sample holder. For pooled samples, the filter area used for TC was 10.4 cm$^2$, slightly exceeding the 9 cm$^2$ limit. Therefore, for TC combustion we used a custom-build quartz spoon, on which up to 16 cm$^2$ of



filter material can be placed and combusted. Filter stacking must be omitted for $^{14}$C(WINSOC) measurement. For this
reason, filter loadings for S1 (pure WINSOC) of the Swiss_4S protocol must be >2 µg C cm$^{-2}$. $^{14}$C(WINSOC) measurements
were omitted in the current study, as only four of the 13 samples had a filter loading >2 µg C cm$^{-2}$ with a mean loading of
1.8 µg C cm$^{-2}$ (range: 0.9–5 µg C cm$^{-2}$).

### 3.4.4 Radiocarbon results

Radiocarbon measurements of TC show a dominant input from fossil carbon in winter months with an average F$^{14}$C of
0.85 ± 0.17 (Table 5). F$^{14}$C values close to non-fossil levels of radiocarbon were found for spring, summer, and autumn with
an average F$^{14}$C of 0.95 ± 0.09 with the highest levels in spring and late summer. Large variations in $^{14}$C(EC) were observed,
ranging from 0.22 to 0.92 (mean: 0.66 ± 0.16). Both the highest and lowest value were observed in winter (23 Feb – 2 Mar
2017 and 23 – 31 Jan 2018), showing that the relative source composition of Arctic carbonaceous aerosol can vary widely
within a season. The highest $^{14}$C(EC) value had the second highest EC concentration (40 ng C m$^{-3}$) and an OC/EC ratio of
5.4, whereas the sample with the very low Fraction Modern carbon had an EC concentration of 16 ng C m$^{-3}$ and OC/EC ratio
of 9.6. Notably, the $^{14}$C(WSOC) content of the high Fraction Modern carbon sample (1.077) was substantially higher than
that of EC indicating different sources of WSOC and EC. Overall, $^{14}$C(WSOC) values showed non-fossil levels of
radiocarbon with maxima in spring and late summer and lower values in early summer and winter. The 31 May to 26 Jun
sample had the lowest $^{14}$C(WSOC) value (0.38), being even lower than the corresponding $^{14}$C(EC) value (0.689), whereas
the calculated value for F$^{14}$C(OC) (0.93) consisted overwhelmingly of carbon from non-fossil sources. Although this might
look contradictory, an explanation can be derived from the concentration of the various fractions. The WSOC concentration
was very low (4 ng C m$^{-3}$), indicating a higher uncertainty, whereas the concentration of WINSOC + EC loss (WINSOC
removal with Swiss_3S) was 93 ng C m$^{-3}$, of which pure WINSOC (S1) accounted for 71 ng C m$^{-3}$. Thus, WINSOC sources
were largely non-fossil.

## 495 4 Conclusions

In the current study, we present an optimised separation procedure for radiocarbon measurements of TC, EC, and WSOC.
Prior to thermal-optical OC/EC separation, a water extraction step was used to minimise charring and to provide eluates for
$^{14}$C(WSOC) measurement. Our method enables radiocarbon source apportionment of the EC and WSOC fraction in addition
to TC, and, when sufficiently loaded filters are available, also the WINSOC fraction. Furthermore, the Fraction Modern of
the OC can be calculated from these values. Prior to AMS $^{14}$C analysis, combustion of TC, EC, and WINSOC are all
performed with a Sunset OC/EC analyser, simplifying the measurement by using a single hyphenation device for multiple
carbonaceous fractions. As demonstrated for the low-loaded Arctic filters, chemical wet oxidation is a simple and reliable
method for measurement of the WSOC fraction, providing low procedural blanks.

We have developed a web tool for calculation of both amount and EC yield, named Sunset-calc, allowing an EC yield
calculation after each run and providing the fraction of charring for each step in the Swiss_3S protocol. Sunset-calc enables
rapid protocol optimisations for a low fraction of charring, while avoiding too large EC losses before the S4 step.

Our thermal desorption model approach for EC yield extrapolation provides a filter-specific non-linear correction based on
the underlying physical properties of the OC/EC mixture and OC composition. The present method supersedes the currently
used linear approach for EC yield extrapolation. Radiocarbon measurements using filters with deliberately lowered EC
yields are no longer necessary. Our approach is independent of season and does not require additional filter material for EC
yield extrapolation, which is crucial when only limited amounts of sample material are available.

**Code availability**

https://github.com/martin-rauber/compycalc
https://github.com/martin-rauber/sunset-calc

**Author contribution**

The work presented here was carried out in collaboration between all authors. S.S. conceived of the study and its design.
M.R. performed the laboratory experiments, implemented the models, and led the preparation of the manuscript. G.S. created
the models and provided guidance and supervision for the laboratory experiments, model implementation, and contributed to
the preparation of the manuscript. K.E.Y. was responsible for collection of the aerosol filter samples and for determining
their OC/EC/TC content. All authors contributed to the editing and proofreading of the manuscript.

**Competing interests**

The authors declare that they have no conflict of interest.

**Acknowledgements**

We would like to thank Jan Strähl for his contribution to the Sunset-calc application and René Bleisch for setting up the R
server. Aerosol filter samples collected at the Zeppelin Observatory, OC/EC/TC and radiocarbon analysis was funded by the
Norwegian Ministry of Climate and Environment.

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

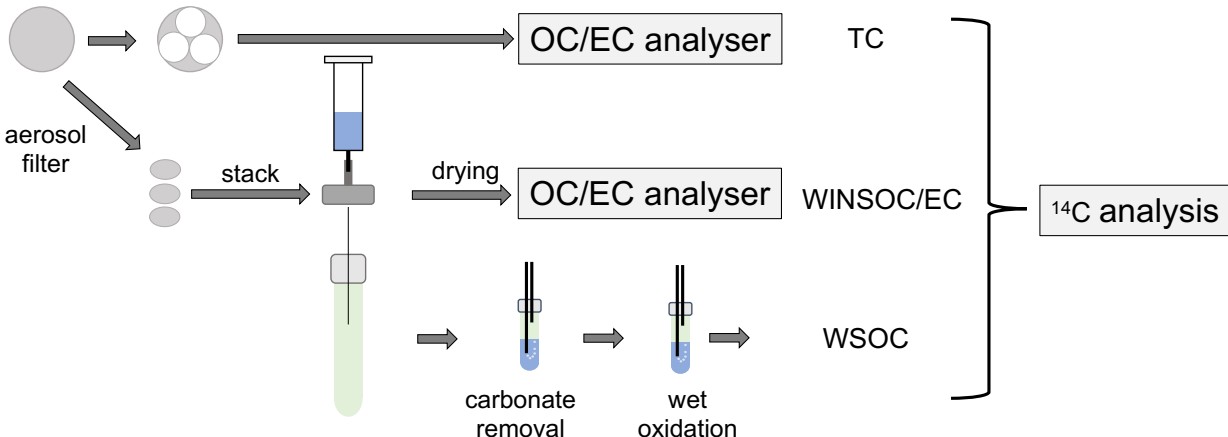

**Figure 1: Separation of the different fractions for ¹⁴C analysis starting from the aerosol filters. One or multiple circular quartz fibre filter punches are stacked and intercalated in the water extraction set-up. The residual filter material used for WINSOC and EC analysis after drying, and the extract oxidised by chemical wet oxidation. The remaining filter material is used for TC analysis.**




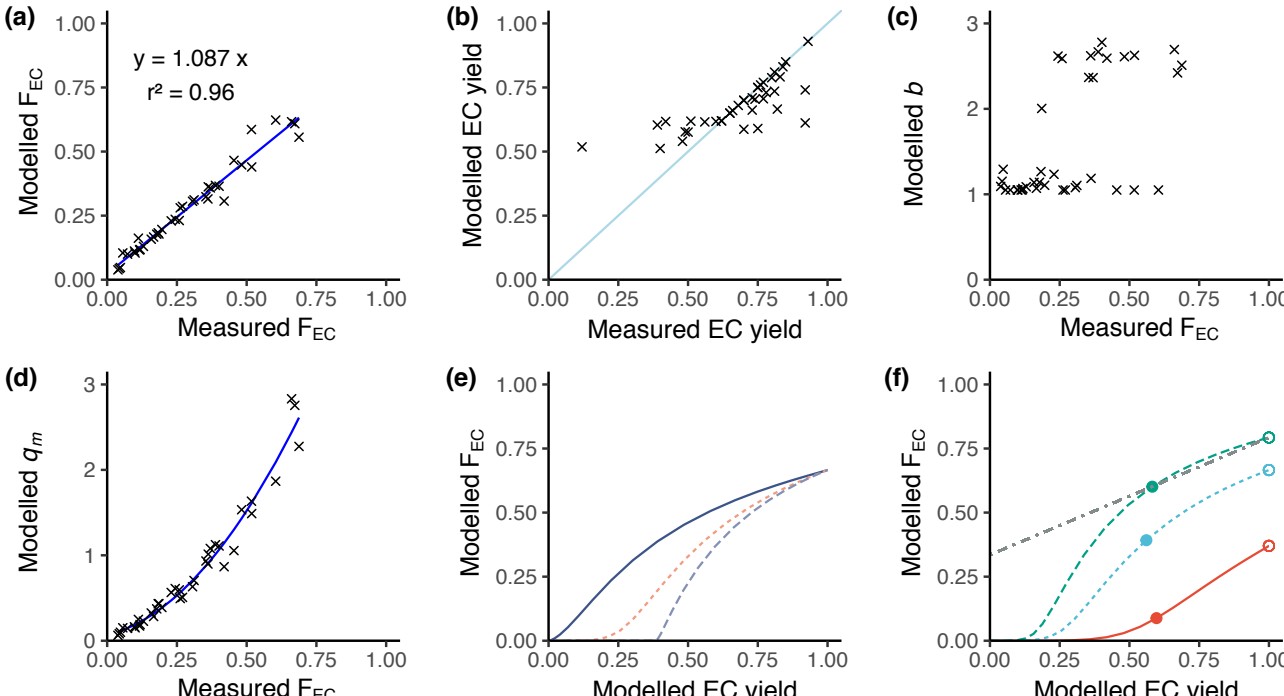

**Figure 2: Summary of the modelled EC correction to an EC yield = 1. a) Model accuracy: modelled $F_{EC}$ vs measured $F_{EC}$. b)**
**Modelled EC yield vs measured EC yield according to Zotter et al. (2014) (see text). c) Model calculated parameters $b$. d) Model**
**calculated parameters $q_m$. e) General behaviour of $F_{EC}$ vs EC yield for different $b$ values (solid line $b = 1.1$, dashed line $b = 1.2$,**
**long-dashed line $b = 1.5$) with a fixed $q_m$ of 1.5. f) General behaviour of $F_{EC}$ vs EC yield for different $q_m$ values (solid line $q_m = 0.5$,**
**dashed line $q_m = 1.5$, long-dashed line $q_m = 2.5$) with a fixed $b$ value of 1.2 and a linear model (dot-dashed line) for a sample with**
**extrapolation at EC yield = 1. Filled dot shows the measured value and the open dots show the value after extrapolation.**






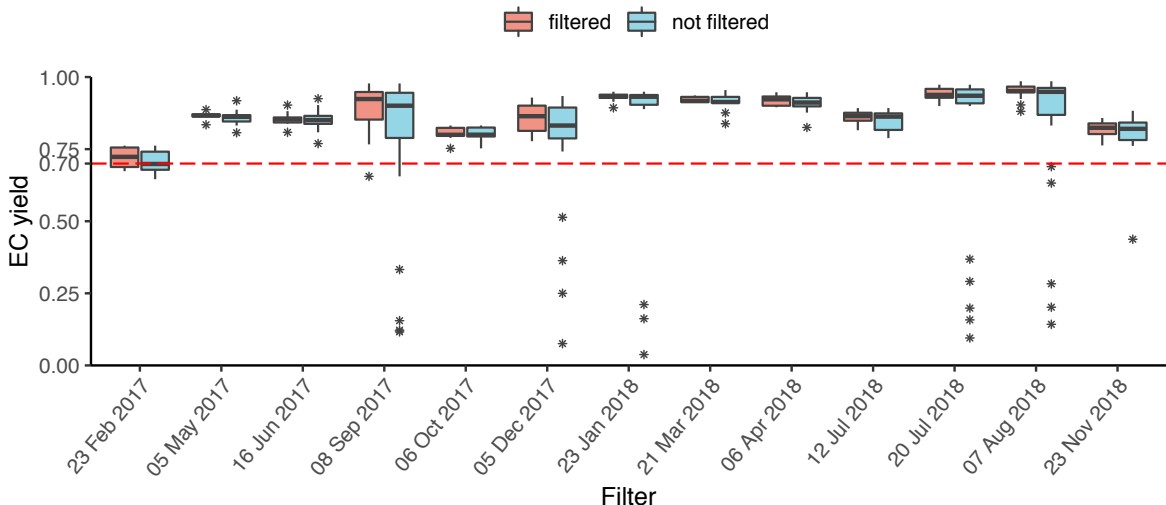

**Figure 3: EC yield after WINSOC removal for each filter with the sampling start date. Filtered (WINSOC removal containing outliers in EC yield, fraction of charring S1, S2, or S3 removed) and unfiltered EC yields for each filter shown. The box plot box shows the first and third quartiles with the mean as a thick horizontal line for the individual groups (filtered and not filtered). The values outside the 3/2 interquartile range are shown with an asterisk. The horizontal line at 0.7 shows that at least 70 % of the initial EC has been recovered.**





Figure 4: Fraction of charring observed for each filter at the individual steps (S1, S2, S3) and the total (sum of S1, S2, S3) with the sampling start date. Filtered (WINSOC removal containing outliers in EC yield, fraction of charring S1, S2, or S3 removed) and unfiltered fractions of charring for each filter shown. The fraction of charring describes the amount of artificially produced EC by charring OC related to the amount of EC on the filter based on the laser transmission signal, i.e., a total charring of 0.05 means a 5 % contamination of the total EC amount.




**Table 1: OC/EC ratios and filter loadings measured by NILU using the EUSAAR_2 protocol. Filters that were pooled for [14]C analysis are marked with an asterisk.**

| Start date | End date | TC ng C m$^{-3}$ | EC ng C m$^{-3}$ | OC ng C m$^{-3}$ | OC/EC ratio |
|---|---|---|---|---|---|
| 23 Feb 2017 | 02 Mar 2017 | 256 | 40 | 216 | 5.4 |
| 05 May 2017 | 15 May 2017 | 158 | 24 | 135 | 5.7 |
| 31 May 2017 | 26 Jun 2017 | 123 | 6 | 117 | 20.5 |
| *08 Sep 2017 | 28 Sep 2017 | 114 | 6 | 108 | 16.7 |
| 28 Sep 2017 | 06 Oct 2017 | 601 | 52 | 549 | 10.5 |
| *06 Oct 2017 | 24 Oct 2017 | 88 | 8 | 81 | 10.4 |
| *05 Dec 2017 | 21 Dec 2017 | 73 | 12 | 61 | 7.7 |
| 23 Jan 2018 | 31 Jan 2018 | 174 | 16 | 157 | 9.6 |
| 21 Mar 2018 | 29 Mar 2018 | 127 | 18 | 109 | 6.1 |
| 06 Apr 2018 | 16 Apr 2018 | 129 | 17 | 111 | 6.4 |
| *12 Jul 2018 | 30 Jul 2018 | 65 | 3 | 62 | 20.7 |
| *30 Jul 2018 | 15 Aug 2018 | 264 | 9 | 254 | 27.0 |
| 23 Nov 2018 | 03 Dec 2018 | 72 | 13 | 59 | 4.5 |
| *Pooled filters | | | | | |

**Table 2: WINSOC amounts for each step of the Swiss_3S protocol measured at the University of Bern and corresponding WSOC**
**amounts. Fraction S1 is considered pure WINSOC, whereas S2 and S3 are mixed fractions of WINSOC and EC. WSOC was determined by subtraction of EC and total WINSOC from TC.**

| Start date | End date | WINSOC (ng C m$^{-3}$) | | | | WSOC ng C m$^{-3}$ | WSOC/WINSOC ratio |
|---|---|---|---|---|---|---|---|
| | | S1 | S2 | S3 | total | | |
| 23 Feb 2017 | 02 Mar 2017 | 43 | 10 | 16 | 70 | 92 | 1.6 |
| 05 May 2017 | 15 May 2017 | 20 | 3 | 8 | 31 | 70 | 2.5 |
| 31 May 2017 | 26 Jun 2017 | 71 | 9 | 12 | 93 | 4 | <0.1 |
| *08 Sep 2017 | 28 Sep 2017 | 13 | 1 | 2 | 16 | 15 | 1.6 |
| 28 Sep 2017 | 06 Oct 2017 | 111 | 26 | 27 | 164 | 284 | 1.9 |
| *06 Oct 2017 | 24 Oct 2017 | 9 | 1 | 2 | 12 | 15 | 1.7 |
| *05 Dec 2017 | 21 Dec 2017 | 13 | 1 | 4 | 18 | 0 | 1.3 |
| 23 Jan 2018 | 31 Jan 2018 | 33 | 5 | 15 | 54 | 59 | 1.1 |
| 21 Mar 2018 | 29 Mar 2018 | 29 | 3 | 5 | 38 | 57 | 1.6 |
| 06 Apr 2018 | 16 Apr 2018 | 26 | 4 | 8 | 37 | 54 | 1.5 |
| *12 Jul 2018 | 30 Jul 2018 | 11 | 0 | 1 | 13 | 7 | 0.7 |
| *30 Jul 2018 | 15 Aug 2018 | 23 | 2 | 3 | 28 | 65 | 2.7 |
| 23 Nov 2018 | 03 Dec 2018 | 22 | 5 | 4 | 32 | 26 | 0.9 |
| *Pooled filters | | | | | | | |



**Table 3: Filter loadings and fractions for front and back filters for TC measured at the University of Bern. n.d. means not determined.**

| Start date | End date | TC front filter ng C m$^{-3}$ | TC back filter ng C m$^{-3}$ | TC$_P$ ng C m$^{-3}$ |
|---|---|---|---|---|
| 23 Feb 2017 | 02 Mar 2017 | 189 | n.d. | n.d. |
| 05 May 2017 | 15 May 2017 | 121 | 28 | 93 |
| 31 May 2017 | 26 Jun 2017 | 113 | 26 | 87 |
| *08 Sep 2017 | 28 Sep 2017 | 39 | 11 | 29 |
| 28 Sep 2017 | 06 Oct 2017 | 501 | 49 | 453 |
| *06 Oct 2017 | 24 Oct 2017 | 35 | 10 | 25 |
| *05 Dec 2017 | 21 Dec 2017 | 36 | 9 | 27 |
| 23 Jan 2018 | 31 Jan 2018 | 135 | 14 | 121 |
| 21 Mar 2018 | 29 Mar 2018 | 109 | 15 | 94 |
| 06 Apr 2018 | 16 Apr 2018 | 105 | 35 | 70 |
| *12 Jul 2018 | 30 Jul 2018 | 26 | n.d. | n.d. |
| *30 Jul 2018 | 15 Aug 2018 | 104 | n.d. | n.d. |
| 23 Nov 2018 | 03 Dec 2018 | 67 | 12 | 54 |

*Pooled filters

**Table 4: Radiocarbon values for EC and OC before (i.e., $F_{EC}$ and $F_{OC}$, respectively) and after the COMPYCALC extrapolation (i.e., $F_{EC(final)}$ and $F_{OC(final)}$, respectively).**

| Start date | End date | $F_{EC}$ F$^{14}$C | $F_{EC(final)}$ F$^{14}$C | $F_{OC}$ F$^{14}$C | $F_{OC(final)}$ F$^{14}$C |
|---|---|---|---|---|---|
| 23 Feb 2017 | 02 Mar 2017 | 0.881 | 0.918 | 0.749 | 0.742 |
| 05 May 2017 | 15 May 2017 | 0.597 | 0.648 | 1.165 | 1.154 |
| 31 May 2017 | 26 Jun 2017 | 0.642 | 0.689 | 0.951 | 0.929 |
| *08 Sep 2017 | 28 Sep 2017 | 0.689 | 0.726 | 0.993 | 0.988 |
| 28 Sep 2017 | 06 Oct 2017 | 0.544 | 0.605 | 1.095 | 1.088 |
| *06 Oct 2017 | 24 Oct 2017 | 0.748 | 0.800 | 0.837 | 0.829 |
| *05 Dec 2017 | 21 Dec 2017 | 0.563 | 0.614 | 0.492 | 0.475 |
| 23 Jan 2018 | 31 Jan 2018 | 0.184 | 0.222 | 0.652 | 0.644 |
| 21 Mar 2018 | 29 Mar 2018 | 0.570 | 0.614 | 1.014 | 1.007 |
| 06 Apr 2018 | 16 Apr 2018 | 0.527 | 0.583 | 1.027 | 1.018 |
| *12 Jul 2018 | 30 Jul 2018 | 0.677 | 0.716 | 0.802 | 0.796 |
| *30 Jul 2018 | 15 Aug 2018 | 0.767 | 0.786 | 1.011 | 1.010 |
| 23 Nov 2018 | 03 Dec 2018 | 0.554 | 0.632 | 0.756 | 0.743 |

*Pooled filters






**Table 5: Final radiocarbon results for each fraction after all calculations and corrections described in this work.**

| Start date | End date | TC $F^{14}C$ | $EC_{final}$ $F^{14}C$ | WSOC $F^{14}C$ | $OC_{final}$ $F^{14}C$ |
|---|---|---|---|---|---|
| 23 Feb 2017 | 02 Mar 2017 | 0.770 | 0.918 | 0.818 | 0.742 |
| 05 May 2017 | 15 May 2017 | 1.068 | 0.648 | 0.987 | 1.154 |
| 31 May 2017 | 26 Jun 2017 | 0.852 | 0.689 | 0.380 | 0.929 |
| *08 Sep 2017 | 28 Sep 2017 | 0.959 | 0.726 | 0.975 | 0.988 |
| 28 Sep 2017 | 06 Oct 2017 | 1.036 | 0.605 | 0.929 | 1.088 |
| *06 Oct 2017 | 24 Oct 2017 | 0.825 | 0.800 | 0.795 | 0.829 |
| *05 Dec 2017 | 21 Dec 2017 | 0.509 | 0.614 | 0.623 | 0.475 |
| 23 Jan 2018 | 31 Jan 2018 | 0.573 | 0.222 | 0.841 | 0.644 |
| 21 Mar 2018 | 29 Mar 2018 | 0.951 | 0.614 | 1.077 | 1.007 |
| 06 Apr 2018 | 16 Apr 2018 | 0.957 | 0.583 | 0.652 | 1.018 |
| *12 Jul 2018 | 30 Jul 2018 | 0.786 | 0.716 | 0.792 | 0.796 |
| *30 Jul 2018 | 15 Aug 2018 | 0.997 | 0.786 | 1.055 | 1.010 |
| 23 Nov 2018 | 03 Dec 2018 | 0.727 | 0.632 | 0.666 | 0.743 |
| *Pooled filters | | | | | |