# Peer review of "An Optimised OC/EC Fraction Separation Method for Radiocarbon Source Apportionment Applied to Low-Loaded Arctic Aerosol Filters"

_EGUsphere, 2022_

## Referee Comment (RC1)

Review of
Rauber et al., *An Optimised OC/EC Fraction Separation Method for Radiocarbon Source Apportionment Applied to Low-Loaded Arctic Aerosol Filters*

The authors present an upgrade on the analytical technique for the measurement of 14C in carbonaceous aerosols. The manner of isolation of EC is a key aspect for conducting radiocarbon-based source apportionment. A plethora of techniques are in use with most employing an intermediate temperature correction step and/or pre-treatment step, while one in particular not employing either correction. Therefore, there is a need for a far superior method. This study addresses this key research gap well.

Yield correction for EC is one the most daunting aspects and major challenges for how $f_m$(EC) results are to be presented/interpreted. The modeling aspect presented here for yield correction is thus critical and something many published works have failed to address or take into account at all. In this regard, the new method is a new benchmark 14C applications for source fingerprinting in atmospheric aerosols.

While the analytical aspect as well as the mathematical aspect are well documented and researched, my main concern is the application to field samples directly. What is the yardstick to know if these numbers from this new method are 'real' values ? Are the readers supposed to take these numbers at face value?

- In the present form the paper is missing an entire section/discussion on reference materials and how well does the present method work for a suite of reference materials. The authors should have addressed this first. In fact, this is a good opportunity to make comparisons with newly established protocols as well (e.g., Huang et al, 2021).

- This relates to my point above, the authors have a good chance to compare with previously published results for 14C -EC analysis for samples collected at Zeppelin observatory (Winiger et al., 2015). While the samples are from 2009 winter and the method used for EC isolation is perhaps the most inferior compared to all out there, the authors could add a discuss section on why a glaring difference in the $f_m$ values is there and if this is related to the yield correction aspect.
My argument here is that based on the method used in Winiger et al, the biomass fraction ought to have been overestimated (as no charring correction or filter pretreatment is done). Even so their $f_m$ values were on average 52% compared to the 66% reported in this study. Why is there this big a difference? Does it mean a bigger input of biomass BC to the Arctic over these years ?
This is precisely my point about which numbers to believe and what could be the 'real' values?

---

## Author Comment (AC1)

Review of
**Rauber et al., An Optimised OC/EC Fraction Separation Method for Radiocarbon**
**Source Apportionment Applied to Low-Loaded Arctic Aerosol Filters**
**(https://doi.org/10.5194/egusphere-2022-625)**

**RC1: Comments by Anonymous Referee #2**

***RC1R: Reply on behalf of all co-authors***

RC1.01

The authors present an upgrade on the analytical technique for the measurement of 14C in carbonaceous aerosols. The manner of isolation of EC is a key aspect for conducting radiocarbon-based source apportionment. A plethora of techniques are in use with most employing an intermediate temperature correction step and/or pre-treatment step, while one in particular not employing either correction. Therefore, there is a need for a far superior method. This study addresses this key research gap well.

Yield correction for EC is one the most daunting aspects and major challenges for how $f_m$(EC) results are to be presented/interpreted. The modeling aspect presented here for yield correction is thus critical and something many published works have failed to address or take into account at all. In this regard, the new method is a new benchmark 14C applications for source fingerprinting in atmospheric aerosols.

*RC1.01R*

*We highly regard the attitude of the Anonymous Referee #2 towards the significance of a comprehensive approach for carbonaceous aerosol analysis. We further thank the reviewer for the appreciation of our efforts to improve the source apportionment of carbonaceous aerosols in general and of EC in particular.*

RC1.02

While the analytical aspect as well as the mathematical aspect are well documented and researched, my main concern is the application to field samples directly. What is the yardstick to know if these numbers from this new method are 'real' values ? Are the readers supposed to take these numbers at face value?

*RC1.02R*

*Although we believe that the theoretic concept based on a first-order kinetic coupled with the Arrhenius equation is superior to all existing approaches to correct for EC losses in thermal-optical separation methods, we agree that the validation of this method is crucial. Unfortunately, this task is not straightforward. Typical validation operations in analytic chemistry employ certified reference materials of high quality that are similar to the sample material of interest and provided by metrological institutions such as the US National Institute of Standards and Technology (NIST). For carbonaceous aerosols on filter material, however, such certified reference materials that may represent ambient condition of PM receptor sites have not been made available yet (see our comment RC1.03R). Alternatively, an agreement of measurement results in intercomparisons provided by different laboratories with expertise in the field is also acceptable. As the correction of EC losses and charring*

*artefacts on $^{14}C$ analyses of EC from PM has not been addressed by other groups to our best knowledge, however, this option is neither available yet. In the work of Zotter et al. (2014), we attempted to fill this gap by investigating, how individual PM filters behave under different conditions of OC removal that lead to varying EC yields, aiming at the development of a correction method that result in identical (within uncertainties) outcomes for the different treatments. Due to the scarcity of the Svalbard filters, however, we could not directly apply the same idea in this work. Nevertheless, we involved the data of Zotter et al. (2014) to establish and optimize the Arrhenius model of this work. This is shown in Figs. 2b as well as S6 and discussed in Chapter 3.1. We consider that this strategy is justified until adequate reference materials are available.*

RC1.03

- In the present form the paper is missing an entire section/discussion on reference materials and how well does the present method work for a suite of reference materials. The authors should have addressed this first. In fact, this is a good opportunity to make comparisons with newly established protocols as well (e.g., Huang et al, 2021).

*RC1.03R*

*Reference materials were not measured, as most of which are provided in powder form only. This powder must be dispersed homogeneously on a filter first, which is difficult to achieve and usually leads to inhomogeneities. Furthermore, such reference materials (e.g., NIST SRM 1649a) typically contain a certain fraction of coarse particles of up to 100 μm, which is substantially larger than the PM10 size cut from the field samples. According to our experience, coarse particles differ in the OC/EC separation and charring behaviour from field samples collected with a PM10 size cut or smaller. We assume that different reaction conditions and kinetics of coarse particles (compared to fine particles) cause these observations: on the one hand, deficient oxygen supply to the interior may entail enhanced pyrolysis; on the other hand, short temperature steps may lead to an inhomogeneous and incomplete heating that hampers chemical reactions. The usefulness of reference materials in powder form is therefore limited. Szidat et al. (2013) utilised NIST SRM 8785 (i.e., SRM 1649a dispersed on filter material using a PM2.5 size cut) for an aerosol intercomparison, however, the study revealed that this material suffers from inhomogeneity, which was caused by the size segregation and filtering step.*

*We have already organized ourselves or participated in intercomparisons between laboratories applying different techniques of EC isolation for $^{14}C$ analysis (Szidat et al., 2013; Zenker et al., 2017), which revealed that employing or omitting water extraction is crucial for an agreement between the individual labs – besides the selection of differing protocols. Most participants in the aerosol intercomparison study from Szidat et al. (2013) did not employ water extraction, which resulted in a larger scatter compared to Zenker et al. (2017), where all participants used water extraction to reduce charring. As the ECT9 protocol employed by Huang et al. (2021) does not include a water extraction step, thus leading to more charring with the present water-soluble OC on the filter, we are unconvinced to apply this protocol ourselves in order to verify our results. Nevertheless, we'd welcome very much, if another intercomparison were organized on the $^{14}C$ analysis of EC from particulate matter (on filters).*

*We thank the reviewer for pointing to the topics of reference materials and intercomparisons and have addressed these in chapter 3.1 in the revised manuscript.*

RC1.04

- This relates to my point above, the authors have a good chance to compare with previously published results for 14C -EC analysis for samples collected at Zeppelin observatory (Winiger et al., 2015). While the samples are from 2009 winter and the method used for EC isolation is perhaps the most inferior compared to all out there, the authors could add a discuss section on why a glaring difference in the $f_m$ values is there and if this is related to the yield correction aspect.

My argument here is that based on the method used in Winiger et al, the biomass fraction ought to have been overestimated (as no charring correction or filter pretreatment is done). Even so their $f_m$ values were on average 52% compared to the 66% reported in this study. Why is there this big a difference? Does it mean a bigger input of biomass BC to the Arctic over these years?

This is precisely my point about which numbers to believe and what could be the 'real' values?

*RC1.04R*

*We think that such a comparison should be taken with caution, as the assumption that the source composition of different winters from the same site are similar may not be correct. For Zeppelin/Svalbard, this has already been proven, as the results from winter 2009 (Winiger et al., 2015) and winters 2012/2013 (Winiger et al., 2019) were substantially different from each other with average fractions of biomass burning ($f_{bb}$) of EC of 0.60 and 0.37, respectively. Nevertheless, we agree that it still makes sense that the results of these studies should be presented here and compared with our data. For this purpose, our results should be converted from the measured $F^{14}C$ into $f_{bb}$ of EC, as the used conversion factors change from year to year (value used for 2017/2018: 1.082). Average $f_{bb}$ of EC for summer (i.e., April – October) and winter (i.e., November – March) amount 0.59 and 0.62, respectively, for this work. There is a good agreement of our summer results with the summer data of Winiger et al. (2019) as well as for our winter results with the winter data of Winiger et al., 2015), whereas there is a large discrepancy of the winter data of Winiger et al. (2019) with both our winter results and the winter data of Winiger et al. (2015). This was discussed in the revised version in Chapter 3.4.4. this way:*

[revised manuscript text omitted]

**RC2: Comments by Anonymous Referee #1**

*RC2R: Reply on behalf of all co-authors*

RC2.01

The paper egusphere-2022-625 by Rauber et al. proposes a new methodology to improve the determination of the fraction of modern carbon for different carbonaceous fractions of atmospheric aerosol.

In the reviewer's opinion the paper is well written, the proposed methodology is extensively discussed, and the topic is of interest for the scientific community. Nevertheless, some weaknesses are present in the manuscript and major revisions are necessary before final publication.

*RC2.01R*

*We thank the Anonymous Referee #1 for acknowledging the relevance of our work for the scientific community. We are further grateful for the efforts of Anonymous Referee #1 to improve our manuscript aiming at a final publication in AMT.*

RC2.02a

Major issue

The most critical aspect concerns the determination of the EC yield required for F(EC). It is determined by comparison of ATN after WINSOC removal and the initial ATN on washed filters.

First of all, differences between optical and thermal quantification of EC by thermal-optical methods has been evidenced in the literature and it should be evidenced and properly discuss, as it strongly impacts the presented results.

Furthermore, some critical aspects concerning the correction approach in the text are not properly discussed:

*RC2.02aR*

*We understand that Anonymous Referee #1 addresses a series of technical details that ought to be tackled in order to improve our approach of an optimum of measured $F^{14}C$ results of EC. Before we discuss the individual suggestions in detail in the following, we'd like to emphasize that our approach aims at a best possible congruence with existing TOA protocols, especially with EUSAAR_2, as these protocols are widely accepted in the community of atmospheric researchers, although they involve several flaws that may be decreased, but probably never will be solved completely. Besides the aspects that Anonymous Referee #1 is pointing to, these flaws of EUSAAR_2, NIOSH and IMPROVE protocols include for instance a) a strict separation of OC and EC irrespective of the fact that there is rather a smooth transition instead of a separation and b) a limitation for the optical correction of charring and determination of the "split point" between OC and EC based on one wavelength only. We are certainly willing to improve our approach even further, however, we cannot eradicate these general flaws of TOA analyses.*

RC2.02b

- pyrolytic carbon (PyC) attenuation coefficient was demonstrated to be different from the EC one (e.g. Yang and Yu, 2002) – and more specifically to be higher than the EC one (e.g. Boparai et al. 2008 and therein cited literature, Subramanian et al., 2006). The authors state that they assume that 50% of the formed PyC evolves in other WINSOC removal steps. Nevertheless, they do not mention the values assumed for the attenuation coefficient of EC and PyC respectively. This information is mandatory to clarify the calculation procedure – and thus the evaluation of EC yield and F(EC) correction. Moreover, as both attenuation coefficients are affected by large uncertainties, an estimate of the impact of these uncertainties on the correction scheme should be evidenced

*RC2.02bR*

*Up to now, we have not considered different attenuation coefficients of EC and pyrolytic carbon (PC). Thanks to the hint in this comment, we implemented this improvement in the revised manuscript (Chapter 2.10, lines 292-298) based on the approach of Winiger et al. (2015) that refers to the work of Chow et al. (2004), which assumes that the actual concentration of PC is only 40% of its apparent value from ATN determination. This approach is also consistent to Boparai et al. (2008). Consequently, a factor of 0.2 (corresponding to 40% of actual PC concentrations related to ATN determination of 50% of the PC that finally may enter the measured EC fraction) is used to correct for both the losses of PC during the thermal treatment and the effect of the different MAC values of PC and EC. All results were recalculated due to this improvement and tables as well as figures were updated accordingly. The implementation of different MAC values for PC and EC changed the absolute values of $F_{EC(final)}$ only marginally by <1% on the average – mainly because the PC contribution was anyway small due to our extraction procedure that was optimized for low PC formation. Estimated uncertainties of $F_{EC(final)}$ and $F_{OC(final)}$ amount ±15% and ±4%, respectively, which was mentioned in the revised manuscript in lines 310-311.*

RC2.02c

-  the lack of a standard reference material for atmospheric EC/BC (already evidenced more than a decade ago – Baumgardner et al., 2012) and of a gold instrumentation still impact all the discussion in the paper. This implies that the "true" value for the yield is unknown. Thus, it should be mentioned in the discussion and conclusions that this method is a step towards improvements in F(EC) determination – but there is currently no real way for a validation of the proposed methodology.

- it is unclear how was PyC quantified. Indeed, PyC formation can be masked by concurrent EC evolution

*RC2.02cR*

*We thank the reviewer for highlighting the potential of our work to the broader community. It is still true that the lack of reference material and an artefact-free instrumentation is missing, which makes it also difficult for our method to be validated properly. The lack of reference materials is already addressed in our reply RC1.03R to a comment of Anonymous Referee #2, which we have implemented in Chapter 3.1 and in the Conclusions in the revised manuscript. We further discuss in our reply RC1.02R that the validation of our method is a crucial task that unfortunately is not straightforward, which we discussed in the revised manuscript also in Chapter 3.1 and the Conclusions.*

*The formation of pyrolytic carbon (PC) was determined using the laser signal/ATN for each step. We typically observed an ATN increase at the moment, when the temperature was increased, whereas to onset of ATN decrease due to EC losses occurred later in each step. This was explained in the revised manuscript in lines 288-290. We cannot exclude, however, that PC formation that may have developed later in the temperature steps was masked by large EC losses. Nevertheless, we regard this as negligible, as the fractions of charring were anyway rather small in the submitted version (see Fig. 4) and became even less relevant by adaptation of comment RC2.02b, which caused a diminution of the weight PC for the determination of $F_{EC}$(final) in general (see RC2.02bR).*

RC2.02d

- evolution of not light-absorbing materials during S1-S3 steps can in principle modify ATN value even if no evolution of light-absorbing components occurs, due to the impact of not light-absorbing materials on transmittance signal related to scattering properties. This effect should be limited by sample washing, but residual effect cannot be excluded. Did the authors evaluate this effect as negligible?

*RC2.02dR*

*Yu et al. (2002) investigated the influence of inorganic compounds on charring and report a certain effect for ammonium bisulfate, which they also assume for similar components such as inorganic nitrate salts. They emphasized the complete water solubility of these compounds and identified water extraction as an efficient removal process. As these inorganic components also constitute the most relevant non-light-absorbing materials, we assume that water extraction also minimizes substantially the effects Anonymous Referee #1 discusses so that we evaluate them as negligible.*

RC2.02e

In the reviewer's opinion, these limitations have to be discussed and insights into the role of these topics on the results have to be gained before manuscript publication. More in detail:

- introduction should be revised evidencing these problems;

- more investigation is needed to revise and improve the discussion in paragraph 3.1 and figure 2. The uncertainty related to the role of PyC in the evaluation of the EC yield and F(EC) correction merits to be evidenced and widely discussed in the text, identifying if it is a major or minor source of uncertainty in the proposed F(EC) correction.

- conclusions have to be extended and the impossibility of a validation of the method related to a lack of a reference material should be evidenced.

*RC2.02eR*

*We followed the suggestions for improvement of Anonymous Referee #1 and implemented them in the revised manuscript at these lines:*

- *lines 68-70 and 88-91*
- *line 351-365; Fig. 2 was updated*
- *lines 288-290*
- *lines 292-298*

- *lines 310-311*

Minor comments

RC2.03

- Introduction: a deeper discussion on the problems concerning EC quantification and isolation should be carried out. More in detail,

*RC2.03R*

*As we already pointed out in RC2.02R (see above), we implemented this discussion in the revised version of the manuscript in the Introduction and further parts.*

RC2.04

- Throughout the paper, "EC measurement", "EC data", … are often used instead of "F(EC) measurement", "F(EC) data", …. This makes the reading quite confusing. Please perform thorough check and modify where needed.

*RC2.04R*

*We checked the manuscript accordingly and corrected terms where necessary.*

RC2.05

Line 104 vs. line 475. Is F(WINSOC) available or not?

*RC2.05R*

*As we already mentioned in Chapters 3.3.2 and 3.4.3, F(WINSOC) was not measured, however with our method it could be measured as well if sufficiently loaded filters are available. We now added a respective comment also in line 104 (line 109 in the revised manuscript).*

RC2.06

Line 128-129: WSOC was not analysed on back filters, whereas TC was because of quantities. Thus, do WINSOC dominate on back filters?

*RC2.06R*

*We only focussed on $^{14}C$ measurements of TC on the back filters in order to provide an estimate of the $^{14}C$ value of the particulate TC fraction, which was determined by an isotopically balanced subtraction of the back from the front filter. As this estimation is already quite uncertain, further measurements on*

*sub-fractions of TC would have been meaningless. As only SVOCs are trapped on the back filters, however, we assume that WSOC fraction is larger than the WINSOC fraction.*

RC2.07

Lines 147-148: was possible adsorption of VOCs on the filter during sample removal and storage quantified?

*RC2.07R*

*VOC adsorption was investigated neither during sample removal nor during storage, as such volatile constituents would anyway have been removed again in the following thermal process.*

RC2.08

Lines 169-171: how was cross contamination quantified? Did the authors test cycles of heating and cooling of zeolites to verify complete CO2 release?

*RC2.08R*

*We are thankful for this comment, which indicates that our description was not comprehensive. The cross contamination was determined in an earlier study (Agrios et al., 2015): After analysing fossil and modern samples alternately, 0.5% of the carbon of the previous sample was found to mix and cross contaminate the next injection. We adapted this passage at the end of Chapter 2.5 in the revised manuscript accordingly.*

RC2.09

Line 262: using ATN to determine EC yield can further complicate the estimate compared to the considerations

*RC2.09R*

*This particular point was already addressed above in RC2.02.R.*

RC2.10

Line 346: TC on back filters look very similar to the values on the front filters. Please check (and, if confirmed, please comment on this).

*RC2.10R*

*This was a mistake. We thank the referee for thorough reading. The correct values are 22 ng C * m$^{-3}$ (range: 12–49), which was put right in the revised manuscript.*

RC2.11

Line 360: why the paragraph on the development of preparation methods is not in section 2?

*RC2.11R*

*It is one of the special features of the journal AMT that both the development and the verification of methods may be part of the outcome that is presented in the Results and Discussion section.*

RC2.12

Line 384-385: this sentence is obscure.

*RC2.12R*

*The sentence was improved as such: The three water-extracted punches from each filter were cut into 12 and 24 quadrants for each individual and pooled sample, respectively.*

RC2.13

Line 409: please change "Al" with "All"

*RC2.13R*

*Corrected in the revised manuscript.*

RC2.14

Line 479-480: how can fossil contribution be dominant if F(TC)=0.85?

*RC2.14R*

*The sentence was improved as such:*

*Radiocarbon measurements of TC show a larger input from fossil carbon in winter months relative to the summer months with an average $F^{14}C$ of $0.85 \pm 0.17$ (Table 5).*

RC2.15

Line 489: are you saying that some fossil sources emit more WSOC than EC compared to non-fossil sources? Any reference for this? If this is not the correct interpretation, what does your result imply?

*RC2.15R*

*We are grateful for this comment, as it uncovered invalid data. Re-evaluating the raw data, we found out that the $^{14}C$ measurements of WSOC from two samples should not have been considered in this study due to a too an insufficient measurement uncertainty that was caused by a too low WSOC amount, one of which concerning the sample from 31 May to 26 Jun discussed in line 489 in the submitted manuscript. In the revised manuscript, the whole passage was removed from Chapter 3.4.4 and Table 5 was updated accordingly.*

---

## Author Response (AR2)

**Review of**
**Rauber et al., An Optimised OC/EC Fraction Separation Method for Radiocarbon**
**Source Apportionment Applied to Low-Loaded Arctic Aerosol Filters**
**(https://doi.org/10.5194/egusphere-2022-625)**

**RC1: Comments by Anonymous Referee #2**

***RC1R: Reply on behalf of all co-authors***

RC1:

Accepted as is.

*RC1.01R*

*We thank the Anonymous Referee #2 for reading the revised manuscript and the positive feedback.*

**RC2: Comments by Anonymous Referee #1**

***RC2R: Reply on behalf of all co-authors***

RC2.01:

Anyway, in the reviewer's opinion, there is still an aspect that must be modified.
The method cannot be considered validated. Thoroughly analysis of high-volume filters performed by the authors was needed for methodology development. But it cannot be claimed as a methodology validation – as no reference material exists, and intercomparison with other methodologies was not carried out.
This must be clearly stated in the conclusions and in par. 3.1 before manuscript publication.

*RC2.01R:*
*We thank the Anonymous Referee #2 for reading the revised manuscript. We agree that the method cannot be seen as fully validated due to the unavailability of suitable reference material. We updated the manuscript and included a clear statement in 3.1 (line 366-367):*

*"Nevertheless, as no suitable reference material exists, the validation of this method is currently not possible and therefore it cannot be considered as fully validated."*

*as well as in the conclusion (line 550-551):*

*"Intercomparison with other methodologies are pending. Furthermore, complete method validation is not feasible due to the unavailability of suitable reference material."*

RC2.02:

- Line 66: "pyrolyses into and forms": please verify

*RC2.02R:*
*We corrected the sentence by removing the word "into" (line 68).*

RC2.03:

- Line 90: "quantification of EC losses and PC formation remain challenging". Please check subject/verb conjugation (singular/plural)
*RC2.03R:*
*We modified and improved the sentence accordingly (line 90-91).*

RC2.04:

- Line 111: the added text: "if this provides sufficient filter loading" is totally obscure to the reviewer.

*RC2.04R:*
*We agree that due to the length of the sentence this may be difficult to read. We changed the wording from "if this provides sufficient filter loading" to "if sufficient filter loading is provided" to be better understandable (line 109).*

RC2.05:

- Line 293-295: another consideration reported in the answers to the reviewers merits to be added: "We cannot exclude, however, that PC formation that may have developed later in the temperature steps was masked by large EC losses. Nevertheless, we regard this as negligible, as the fractions of charring were anyway rather small

*RC2.05R:*
*We thank the reviewer for this remark and added this missing information into the manuscript (line 290-292).*

RC2.06:

- lines 519-536: independently of the number of samples, what is the time-covering of the periods considered in the different years? Are seasons suitably represented by the analysed samples (e.g. how many days per season are represented in the analysed samples)? If not, all the comparison among different years is nonsense.

*RC2.06R:*
*The time covered overall is 175 days, of which 105 were sampled in 2017 and 70 in 2018. Overall, 49 days were covered in winter versus 126 days for summer. Although the periods covered between the seasons are different, we do not consider this time-covering inconsistent with the comparisons performed in this work.*